# Selective Photothermal Therapy Using Antioxidant Nanoparticles Encapsulating Novel Near-Infrared-Absorbing Platinum(II) Complexes

**DOI:** 10.3390/nano15110796

**Published:** 2025-05-25

**Authors:** Ryota Sawamura, Hiromi Kurokawa, Atsushi Taninaka, Takuto Toriumi, Yukio Nagasaki, Hidemi Shigekawa, Hirofumi Matsui, Nobuhiko Iki

**Affiliations:** 1Graduate School of Environmental Studies, Tohoku University, 6-6-07 Aramaki-Aoba, Aoba-ku, Sendai 980-8579, Miyagi, Japan; 2Phycochemy Corporation, X/S Worksite, 4-19-1 Midorigahara, Tsukuba 305-0035, Ibaraki, Japan; hkurokawa.tt@md.tsukuba.ac.jp; 3Faculty of Medicine, University of Tsukuba, 1-1-1 Tennodai, Tsukuba 305-8575, Ibaraki, Japan; hmatsui@md.tsukuba.ac.jp; 4Institute of Pure and Applied Sciences, University of Tsukuba, 1-1-1 Tennodai, Tsukuba 305-8573, Ibaraki, Japan; jun_t@bk.tsukuba.ac.jp (A.T.); hidemi@bk.tsukuba.ac.jp (H.S.); 5Takano Co., Ltd., 137 Miyada-mura, Kamiina-gun 399-4301, Nagano, Japan; 6Faculty of Materials for Energy, Shimane University, 1060 Nishikawatsu-cho, Matsue 690-8504, Shimane, Japan; ttoriumi@mat.shimane-u.ac.jp; 7Graduate School of Pure and Applied Sciences, University of Tsukuba, 1-1-1 Tennodai, Tsukuba 305-8575, Ibaraki, Japan; nagasaki@ims.tsukuba.ac.jp; 8Center of Applied Nanomedicine, National Cheng Kung University, No. 35, Xiaodong Rd., Tainan 701, Taiwan

**Keywords:** photothermal therapy, near-infrared absorption, diradical-platinum complexes, antioxidant nanoparticles, cancer cell-specificity

## Abstract

Photothermal therapy (PTT) is a promising approach for cancer treatment that has minimal side effects. It locally heats tumors using agents that convert near-infrared (NIR) light energy into heat. We previously reported that the NIR-absorbing hydrophobic diradical-platinum(II) complex PtL_2_ (L = 3,5-dibromo-1,2-diiminobenzosemiquinonato radical) can kill cancer cells through its photothermal conversion ability. In this study, we developed PtL_2_-loading nanoparticles (PtL_2_@RNPs) for the delivery of the complex to tumors based on the enhanced permeability and retention effect using an amphiphilic block copolymer that can scavenge reactive oxygen species. PtL_2_@RNPs exhibited particle diameters of 20–30 nm, an encapsulation efficiency exceeding 90%, and loading capacities of up to 12%. Under NIR laser irradiation, PtL_2_@RNPs stably generated heat with almost 100% photothermal conversion efficiency. Although the particles were not modified for cancer cell targeting, their uptake by cancer cells was approximately double that by normal cells. PtL_2_@RNPs exhibited NIR absorption and effectively killed cancer cells at a low irradiation power (0.15 W). Normal cells treated with PtL_2_@RNPs remained largely undamaged under identical irradiation conditions, demonstrating a cancer-cell-specific photothermal killing effect. These findings can provide insights for future basic studies on cancer cells and the development of effective cancer treatment modalities.

## 1. Introduction

Photothermal therapy (PTT) has attracted research attention in the field of cancer treatment [1,2,3]. It utilizes photothermal agents (PTAs) that convert the energy of near-infrared (NIR) light into heat. In the NIR region (700–1100 nm), endogenous substances (e.g., water and hemoglobin) do not exhibit strong absorption [4]. Moreover, the energy of NIR photons is lower than that of radioactive rays (X-rays, γ-rays, etc.) used for radiotherapy. Consequently, NIR light can penetrate biological tissue with minimal invasiveness and interference. PTAs that enter tumor sites heat them using NIR light irradiation from outside the body. When the intracellular temperature exceeds 43 °C, cell death is induced [5,6]. In addition to initiating therapeutic effects, photoirradiation facilitates the precise spatial control of heating confined to the vicinity of PTAs. If PTAs themselves are not toxic, they may reduce unintended side effects compared with traditional anticancer drugs. Therefore, PTT is expected to provide effective cancer treatment with minimal damage to healthy tissue.

Conventional PTT studies have adopted various NIR-absorbing materials. Gold nanorods exhibit NIR absorption due to localized surface plasmon resonance and efficient photothermal conversion [7,8]. Carbon nanotubes and graphene oxide also exhibit photothermal conversion upon NIR irradiation [9,10]. Their rigid structures enable chemical modification of their surfaces to improve biocompatibility and enhance cancer cell targeting [11,12,13]. However, there are concerns regarding inflammation or fibrosis due to long-term retention in the body [14,15]. Cyanine dyes are less toxic than inorganic materials and have been used for NIR fluorescence imaging [16,17]. They can convert the energy of NIR light into heat, but their photothermal conversion efficiencies (*η*) are typically less than 50% [18,19]. Photodegradation of cyanine dyes is another problem [18]. The development of PTAs with balanced biocompatibility and photophysical and photothermal properties is crucial for improving the therapeutic efficacy of PTT.

To achieve cancer-specific treatment with minimal side effects, PTAs should be delivered to tumor sites accurately and safely. For PTT using small PTAs, the agents have been encapsulated into nanocarriers in many studies as a promising strategy for controlled drug delivery [20,21,22]. In malignant tissue, blood vessels become leaky owing to active angiogenesis, and immature lymph ducts lack the ability to excrete waste substances. Based on these physical characteristics, nanosized materials with a diameter of 10–200 nm can easily permeate blood vessels and accumulate in tumor tissue; this is known as the enhanced permeability and retention (EPR) effect [23]. Among the various types of nanocarriers, polymeric micelles composed of amphiphilic polymers have advantages such as ease of preparation, monodisperse size distribution, lack of surface modification, and efficient solubilization of hydrophobic drugs [24].

Our group previously studied diradical-platinum(II) complexes composed of Pt^II^ ions and two *o*-diiminobenzosemiquinonato radical ligands as candidate NIR-absorbing materials for PTT. These complexes exhibit an absorption peak in the NIR range (700–800 nm) with molar absorption coefficients (*ε*) of 10^4^–10^5^ M^−1^ cm^−1^ through ligand-to-ligand charge transfer (LLCT) [25]. We previously reported that the hydrophobic complex PtL_2_ (Figure 1, L = 3,5-dibromo-1,2-diiminobenzosemiquinonato radical) exhibited stable photothermal conversion without photodegradation and killed cancer cells via its photothermal effect during NIR laser irradiation [26]. However, PtL_2_ showed higher cytotoxicity than cisplatin (an anticancer drug), which may have been caused by intracellular crystallization [26,27]. Considering that PtL_2_ does not have cancer-targeting ability, a delivery system is needed to safely deliver the complex to tumor sites.

In this study, we developed PtL_2_-loading antioxidant nanoparticles PtL_2_@RNPs (Figure 1). The amphiphilic copolymer PEG-*b*-PMNT consists of a hydrophilic polyethylene glycol (PEG) segment and a hydrophobic PMNT segment, in which 2,2,6,6-tetramethylpiperidine-N-oxyl (TEMPO) moieties are introduced as side chains onto the poly(methylstyrene) backbone via amine linkers [28,29]. This copolymer in an aqueous solution forms nanomicelles, which are named redox nanoparticles (RNPs). When RNPs disintegrate, TEMPO radicals scavenge the reactive oxygen species (ROS) that influence gene expression near tumors and promote malignancy and metastasis. Combining RNPs with other therapeutic modalities can improve treatment efficacy while preventing side effects [30]. Therefore, PtL_2_@RNPs are expected to enhance the PTT efficacy of PtL_2_ by eliminating ROS in PEG-*b*-PMNT polymers. Herein, we report the photophysical properties and photothermal cell killing ability of PtL_2_@RNPs. These nanoparticles exhibited particle sizes suited for the EPR effect and strong NIR absorption. The *η* value of PtL_2_@RNPs was almost 100%, indicating highly efficient photothermal conversion. Experiments were performed using rat gastric mucosal cells (RGM1) and cancerous mutant cells (RGK1) of the same genetic origin [31,32]. Although PtL_2_@RNPs did not recognize cancer cells, they were introduced more frequently into RGK1 cells than into RGM1 cells. The nanoparticles in the cells also showed NIR absorption, resulting in photothermal damage to the RGK1 cells. In contrast, the RGM1 cells containing PtL_2_@RNPs remained mostly unaffected, even at the same irradiation power. Therefore, PtL_2_@RNPs exhibited a cancer-cell-specific killing effect via their photothermal conversion capabilities.

## 2. Materials and Methods

### 2.1. Reagents

Potassium tetrachloroplatinate(II) (K_2_PtCl_4_), hydrochloric acid (HCl), nitric acid (HNO_3_), *N*,*N*-dimethylformamide (DMF), dimethyl sulfoxide (DMSO), sodium chloride (NaCl), potassium chloride (KCl), sodium hydrogen phosphate (Na_2_HPO_4_), and potassium dihydrogen phosphate (KH_2_PO_4_) were purchased from Kanto Chemical Co., Inc. (Tokyo, Japan). 3,5-Dibromo-1,2-phenylenediamine monohydrochloride (H_2_L·HCl) was supplied from Tokyo Chemical Industry Co., Ltd. (Tokyo, Japan). Nitric acid (HNO_3_), hydrogen peroxide (H_2_O_2_) solution, 2-[4-(2-hydroxyethyl)piperazin-1-yl]ethanesulfonic acid (HEPES), magnesium chloride hexahydrate (MgCl_2_·6H_2_O), calcium chloride dihydrate (CaCl_2_·2H_2_O), and L-glutamine were obtained from FUJIFILM Wako Pure Chemical Corporation (Osaka, Japan). Glucose was obtained from Sigma-Aldrich. Dulbecco’s modified Eagles medium/Ham’s F-12 (DMEM/F-12), fetal bovine serum (FBS), penicillin/streptomycin (PS), and trypsin/ethylenediaminetetraacetic acid (EDTA), were purchased from Thermo Fisher Scientific (Waltham, MA, USA). Cell Counting Kit-8 (CCK-8) and Hoechst 33342 were supplied from DOJINDO LABORATORIES (Kumamoto, Japan). All reagents were used without further purification.

### 2.2. Instruments

Proton nuclear magnetic resonance (^1^H NMR) spectrum was measured with a Bruker DPX400 spectrometer. Elemental analysis was performed by using an elemental analyzer JM-11 (J-SCIENCE LAB Co., Ltd., Kyoto, Japan). Z-contrast images of PtL_2_@RNPs were taken by a HD-2700 scanning transmission electron microscope (Hitachi High-Tech, Tokyo, Japan). The nanoparticles were loaded on a Collodion film-coated TEM grid (Nissin EM Co., Ltd., Tokyo, Japan) and dried overnight before observation. Ultraviolet-visible (UV-Vis) absorption spectra were recorded by a UV-1800 spectrometer (SHIMADZU CORPORATION, Kyoto, Japan). Dynamic light scattering (DLS) and zeta potential measurements were performed by a nanoPartica SZ-100 nanoparticle analyzer (HORIBA, Ltd., Kyoto, Japan). The absorbance of each well in a 96-well plate was measured using an iMark™ microplate absorbance reader (Bio-Rad Laboratories, Hercules, CA, USA). Inductively Coupled Plasma Atomic Emission Spectroscopy (ICP-AES) measurement was performed by Thermo iCAP 6500 spectrometer (Thermo Fisher Scientific, Waltham, MA, USA).

### 2.3. Synthesis of PtL_2_

Complex PtL_2_ was synthesized based on the reported procedure [33]. The HCl solution (0.01 M, 9 mL) of K_2_PtCl_4_ (0.031 g, 0.075 mmol) and H_2_L·HCl (0.046 g, 0.150 mmol) was refluxed with stirring at 70 °C for 24 h under argon atmosphere. Water/DMF (1:1 *v*/*v*, 18 mL) was then added to the mixture, and the insoluble substances were filtered. The obtained filtrate was heated with shielding light at 50 °C for 2 days. The blue–violet precipitates were collected by filtration, washed with water/DMF (1:2, 1:1 *v*/*v*) and water, and dried in vacuo (0.023 g, yield 43%). PtL_2_ was used as the mixture of *cis* and *trans* isomers. ^1^H NMR (400 MHz, DMSO-*d*_6_): *δ* 7.21/7.22 (dd, *J*_1_ = 1.60/2.00 Hz, *J*_2_ = 4.40/4.80 Hz, 1H, Ar*H*), 7.42 (br, 1H, Ar*H*). C_12_H_8_N_4_Br_4_Pt (722.92): calcd. C 19.94, H 1.12, N 7.75; found C 20.02, H 1.25, N 7.66.

### 2.4. Preparation of PtL_2_@RNPs

The PEG-*b*-PMNT polymer (PEG *M*_n_ 5000, PMNT *M*_n_ 3200, *m* = 114, *n* = 19) was prepared according to a previous report [28]. Complex PtL_2_ and PEG-*b*-PMNT were dissolved in DMF (1 mL) with a mass ratio of 1:10, 1:20, and 1:40. After mixing for 1 h, pure water (1 mL) was gradually added to the DMF solution. The resulting mixture was dialyzed against water for 24 h. The outer solution of the dialysis membrane was exchanged for fresh pure water after 2, 6, and 20 h. Finally, we collected the inner solution as an aqueous solution of PtL_2_@RNPs.

The Pt^II^ concentration of the obtained solution was determined by absorption spectrometry. An aqueous PtL_2_@RNPs (0.1 mL) was evaporated and then dissolved in DMF (5 mL) using a volumetric flask. The absorbance at 732 nm of the resulting DMF solution was measured to calculate the Pt^II^ concentration based on the *ε*_732_ of PtL_2_ in DMF (1.1 × 10^5^ M^−1^ cm^−1^). The obtained aqueous solution of PtL_2_@RNPs was used after the solvent was exchanged for PBS, as needed.

### 2.5. Solution Temperature Measurement During NIR Laser Irradiation

The experimental system was set up according to the reported method [8]. The PBS solution of PtL_2_@RNPs (PtL_2_:PEG-*b*-PMNT = 1:20 *w*/*w*, 2 mL) was put in a 1 cm quartz cuvette and stirred at 300 rpm during the experiment. A magnetic stirrer was set 5 cm apart from the bottom of the cuvette. The cuvette was capped by Styrofoam, having a small hole for a thermocouple to measure the solution temperature. This is to restrict heat transfer to that through the cuvette walls. The NIR semiconductor laser (730 nm, NaKu Technology Co., Ltd., Hangzhou, Zhejiang, China) was used with a collimated optical fiber attached. The laser power (2 W cm^−2^) was calibrated at an approximate median considering variations in the detected value by the laser power meter (PM160T-HP, Thorlabs, Inc., Newton, NJ, USA). The thermocouple probe was carefully fixed so as not to disturb the light path of the NIR laser. The laser beam was focused with a plano-convex lens to set the spot size to 5 mm in diameter.

### 2.6. Cell Culture

RGM1 cells were cultivated in DMEM/F-12 medium supplemented with L-glutamine. RGK1 cells were cultured in DMEM/F-12 medium without L-glutamine. Both culture media contained 10% FBS and 1% PS. Cells were cultured at 37 °C under the humidified atmosphere of 5% CO_2_ and passaged when reached 70–80% confluency.

### 2.7. Cytotoxicity Assay

RGM1 or RGK1 cells (5.0 × 10^3^ cells) were seeded in clear bottom 96-well plates and preincubated at 37 °C overnight. After the culture media were removed, the media containing PtL_2_@RNPs (PtL_2_:PEG-*b*-PMNT = 1:20 *w*/*w*) with different Pt^II^ concentrations (0–4.0 × 10^−5^ M) were poured into the wells and then incubated at 37 °C for 24 h. The media containing PtL_2_@RNPs were replaced by CCK-8 diluted by a fresh culture medium. After the incubation for 1 h, the absorbance at 450 nm in each well was measured by a microplate reader.

### 2.8. Hyperspectral Imaging of Cells Containing PtL_2_@RNPs

Hyperspectral imaging was performed using the hyperspectral camera NH-1TIK (EBA JAPAN Co., Ltd., Tokyo, Japan) attached to the OLYMPUS polarization microscope BX51 with a liquid immersion objective lens (40×). This camera can take sub-images in the wavelength range of 400–1000 nm at 5 nm intervals. The acquisition data were analyzed and converted to images by HSAnalyzer software (Ver. 1.0, EBA JAPAN Co., Ltd., Tokyo, Japan). Color images were constructed by sub-images at 480, 545, and 700 nm as blue, green, and red, respectively. The absorption spectrum of the region of interest (ROI) was calculated from the signal intensity at each wavelength of ROI *I*(*λ*) divided by that of areas without cells *I*_BG_(*λ*), according to Lambert–Beer’s law. It is noted that the signal intensity *I* is the cumulative value of all pixels in the area.(1)$$A\left(\lambda \right)=-\log_{10}\frac{I\left(\lambda \right)}{I_{\mathrm{BG}}\left(\lambda \right)}$$

RGM1 or RGK1 cells (1.0 × 10^5^ cells) were seeded in 35-mm culture dishes and then preincubated at 37 °C overnight. The media were replaced by PtL_2_@RNPs (PtL_2_:PEG-*b*-PMNT = 1:20 *w*/*w*)-containing media followed by the incubation at 37 °C for 24 h. After the media were removed, the dishes were rinsed with PBS, and MSF buffer was poured into the dishes. The MSF buffer is composed of NaCl (1.37 × 10^−1^ M), KCl (5.4 × 10^−3^ M), Na_2_HPO_4_ (3.3 × 10^−4^ M), KH_2_PO_4_ (4.4 × 10^−4^ M), HEPES (1.01 × 10^−2^ M), glucose (8.3 × 10^−3^ M), MgCl_2_·6H_2_O (1 × 10^−3^ M), and CaCl_2_·2H_2_O (1 × 10^−3^ M). Finally, the cells were observed by a microscope equipped with NH-1TIK.

### 2.9. Observation for Intracellular Distribution of PtL_2_@RNP by Spectral Angle Mapper Algorithm

The spectral angle mapper algorithm is a method to classify the pixels by spectrum similarity with the reference spectrum [34]. When the spectra of ROI and reference are described as vectors ***t*** and ***r***, the spectral similarity can be written as the angle between the two vectors:(2)$$\theta =\cos^{-1}\left(\frac{t\cdot r}{\left\|t\right\|\cdot \left\|r\right\|}\right)$$

In the distribution maps, the pixels having *θ* smaller than a threshold angle were colored in white, and those having *θ* larger than a threshold were colored in black.

RGM1 or RGK1 cells (1.0 × 10^5^ cells) were seeded in 35 mm culture dishes and then preincubated at 37 °C overnight. The media were replaced by PtL_2_@RNPs-containing media followed by the incubation at 37 °C for 24 h. After removing the media, the dishes were rinsed with PBS, and Hoechst 33342 diluted by MSF buffer was poured into the dishes. The cells were incubated at room temperature in the dark for 5 min and washed with PBS. After fresh MSF buffer was poured into the dishes, the cells were observed by a fluorescent microscope equipped with NH-1TIK.

### 2.10. Quantitative Analysis of the Intracellular Amount of Pt^II^ Ions

RGM1 or RGK1 cells (1.0 × 10^5^ cells) in culture dishes were incubated with PtL_2_@RNPs (PtL_2_:PEG-*b*-PMNT = 1:20 *w*/*w*, [Pt^II^] = 2.0 × 10^−5^ M) for 24 h. The cells were peeled off from the dishes and suspended in PBS. The cell suspensions were mixed with concentrated HNO_3_ (1 mL) and H_2_O_2_ solution (1 mL) and then boiled at 95 °C to dry up completely. The residue was dissolved in HNO_3_ (final concentration: 0.1 M) and filled to 5 mL with pure water using a volumetric flask. The Pt^II^ concentration in the sample measured by ICP-AES was divided by the number of cells in the used suspension to afford the intracellular amount of Pt^II^ ions (mol cell^−1^).

### 2.11. Time-Lapse Observation of Cell-Killing by the Photothermal Conversion of PtL_2_@RNPs

RGM1 and RGK1 cells were grown overnight in a 35-mm film-base dish (Matsunami Glass Ind., Ltd., Osaka, Japan) and then incubated with PtL_2_@RNPs (PtL_2_:PEG-*b*-PMNT = 1:20 *w*/*w*, [Pt^II^] = 2.0 × 10^−5^ M) for 24 h. After incubation, the medium was changed to the fresh medium of 0.5 mL, an optical fiber with a cladding diameter of 125 µm was placed in contact with the bottom of the dish, and cells were irradiated with 730 nm NIR laser (0.28, 0.15, and 0.086 W) for 10 min. After irradiation, phase-contrast and time-lapse observations were performed with an IX83 microscope system (Olympus Corp., Tokyo, Japan) equipped with a stage-top incubator (Tokai Hit Co., Ltd., Shizuoka, Japan).

## 3. Results and Discussion

### 3.1. Characterization of PtL_2_@RNPs

Figure 2a shows the particle size distributions of the free RNPs and PtL_2_@RNPs prepared using different PtL_2_/PEG-*b*-PMNT mass ratios. Their mean hydrodynamic diameters and zeta potentials are summarized in Table 1. The particle size under all conditions was 20–30 nm, which is suitable for drug delivery based on the EPR effect. Z-contrast images of PtL_2_@RNPs also exhibited spherical particles with a diameter of ca. 20 nm (Appendix A). The zeta potentials were almost 0 mV, indicating that the amphiphilic copolymer PEG-*b*-PMNT formed nanosized micelles in which the outer shell consisted of PEG chains and the inner core consisted of PMNT chains containing PtL_2_. The encapsulation efficiency (EE) and loading capacity (LC) of PtL_2_@RNPs were calculated using the following equations:(3)$$\mathrm{EE}=\frac{\mathrm{Mass}\;\mathrm{of}\;{\mathrm{PtL}}_{2}\;\mathrm{in}\;\mathrm{RNPs}\;[g]}{\mathrm{Feeding}\;\mathrm{mass}\;\mathrm{of}\;{\mathrm{PtL}}_{2}\;[g]},$$(4)$$\mathrm{LC}=\frac{M\mathrm{ass}\;\mathrm{of}\;{\mathrm{PtL}}_{2}\;\mathrm{in}\;\mathrm{RNPs}\;[g]}{\mathrm{Total}\;\mathrm{mass}\;\mathrm{of}\;{\mathrm{PtL}}_{2}@\mathrm{RNPs}\;[g]}.$$

These values are listed in Table 1. The three conditions resulted in an EE exceeding 90%. These LCs increased with an increasing mass fraction of PtL_2_. For reference, the PEG-poly(L-lactic acid) (PEG-PLLA) multiblock copolymer (*M*_n_ > 35,000) encapsulated hydroxycamptothecin (anticancer drug) at 80% EE and 11.7% LC, forming micelles with diameters of 90.3 nm [35]. A fluorescent dye-loaded PEG-poly(ε-caprolactone)-polyethyleneimine (PEG-PCL-PEI) triblock copolymer micelle, which has an average diameter of 27.74 nm, shows 75.37% EE and 3.47% LC in a dye/polymer mass ratio of 1:20 [36]. Compared to these nanomicelles, PtL_2_@RNPs achieved a higher EE and LC despite the presence of small particles, indicating that RNPs are suitable carriers for the delivery of PtL_2_.

The UV–Vis absorption spectrum of PtL_2_@RNPs in PBS showed a characteristic absorption peak at 600–850 nm with a shape similar to that of PtL_2_ in DMF derived from the LLCT band (Figure 2b). The maximum absorption wavelength of PtL_2_@RNPs shifted by 10 nm from that of PtL_2_ in DMF. Our group has reported that inclusion into β-cyclodextrin causes redshifts of absorption wavelengths for hydrophilic complexes [37,38]. This is likely due to the limited solvent accessibility to the complex. Similarly, in the case of PtL_2_@RNPs, encapsulation in RNPs may have caused spectroscopic changes. As the mass fraction of PtL_2_ increased, the *ε* value at 742 nm decreased (7.8 × 10^4^ M^−1^ cm^−1^ (PtL_2_/PEG-*b*-PMNT 1:40 *w*/*w*), 6.9 × 10^4^ M^−1^ cm^−1^ (1:20), and 5.5 × 10^4^ M^−1^ cm^−1^ (1:10)). In contrast, the *ε* values at 620–680 nm increased. Here, the aggregation of PtL_2_ in RNPs can be ruled out because PtL_2_ aggregates exhibit broad absorption at approximately 550 nm [27]. As mentioned above, the LCs increased in response to the mass ratio of PtL_2_, whereas the particle sizes remained almost identical. Therefore, the hydrophobic interactions between PtL_2_ and PMNT chains are thought to strengthen at higher PtL_2_ concentrations. This may have influenced the transition dipole moment of each vibrational energy level in the LLCT band. Considering the balance of the LC and *ε* values, we used PtL_2_@RNPs prepared using a PtL_2_/PEG-*b*-PMNT mass ratio of 1:20 in subsequent experiments.

To evaluate long-term storage stability, we stored the prepared PBS solution of PtL_2_@RNPs in a refrigerator for one month and then measured its physical and spectrometric properties. Although the mean diameter increased slightly by 1.2 nm after one month, the particle size distribution remained almost unchanged (Appendix A). The zeta potential remained at approximately zero. Although the absorbance at 742 nm decreased by 7.6% (Appendix A), no broad absorption peaks derived from the aggregates were observed. Moreover, no precipitates were observed in the solution. These results imply that the RNPs can stably encapsulate PtL_2_ for long periods.

### 3.2. Photothermal Conversion Properties of PtL_2_@RNPs

Next, the temperature elevation of the PBS solutions of PtL_2_@RNPs with different Pt^II^ concentrations was measured during irradiation with an NIR laser (laser power density: 2 W cm^−2^) for 30 min. The rise in solution temperature from the initial value (Δ*T*) increased in response to the Pt^II^ concentration (Figure 3a). The temperature elevation became broadly similar in conditions more than 2.0 × 10^−5^ M because the light attenuation rate 1–10^−^*^Aλ^* approximates to 1. The Δ*T* value after 30 min reached 15.3 °C at the highest concentration (4.0 × 10^−5^ M); in contrast, the temperature increase for PBS alone was only 0.4 °C. The absorption spectra before and after NIR irradiation did not change for any Pt^II^ concentration (Appendix A). Furthermore, PtL_2_@RNPs in PBS exhibited similar temperature curves for five repeated cycles of irradiation (Figure 3b). Spectral changes in the nanomicelles were not observed in the cycle irradiation experiment (Appendix A). Furthermore, the absorbance at 742 nm of PtL_2_@RNPs heated above 45 °C for 4 h decreased by only 5–7% (Appendix A) and no precipitates were observed. PtL_2_@RNPs were stable against NIR irradiation and temperature elevation. The *η* value of PtL_2_@RNPs was calculated as 99.9% (Appendix A), almost the same as that of PtL_2_ (95.4%, Appendix A). Gold nanorods used as a reference for calculations in the literature exhibited efficiencies of over 95% [8]. In addition, the *η* value of PtL_2_@RNPs exceeded those of reported PTAs, such as polymeric nanomicelles encapsulated acceptor–donor–acceptor structure molecules (31.5%) [19], cyanine dye-loaded mesoporous polymeric nanoparticles (54.92%) [21], and albumin-coated ultrasmall metallic nanodots (43.99%) [39]. The combination of nonemissive PtL_2_ and thermally stable RNPs may have resulted in highly efficient photothermal conversion.

### 3.3. Cytotoxicity of PtL_2_@RNPs for RGM1 and RGK1 Cells Under No NIR Irradiation

Figure 4 shows the viability of RGM1 (normal) and RGK1 (cancer) cells after incubation with PtL_2_@RNPs at different Pt^II^ concentrations (0–4.0 × 10^−5^ M) for 24 h. Both cell lines survived approximately 100% under all conditions. In contrast, the half-maximal inhibitory concentration (IC_50_) of PtL_2_ against human breast cancer MCF-7 cells was comparable to that of the anticancer drug cisplatin [26]. We considered that the dark cytotoxicity of PtL_2_ was due to intracellular crystallization of hydrophobic PtL_2_ [27]. The encapsulation of PtL_2_ into RNPs can significantly reduce its toxicity and facilitate safe transformation to the target sites.

### 3.4. Difference in the Uptake of PtL_2_@RNPs for Normal and Cancer Cells

Next, we observed RGM1 and RGK1 cells after incubation with PtL_2_@RNPs ([Pt^II^] = 4.0 × 10^−5^ M) for 24 h through hyperspectral imaging. This imaging method combines microscopic techniques with spectrometry and provides images containing spectral information. Color images indicated that more blue vesicles were present in RGK1 cells than in RGM1 cells (Figure 5a,b). The absorption spectrum of the black-framed region in Figure 5b resembles that of PtL_2_@RNPs in PBS (Figure 5c). This implies that the blue vesicles in the cells contained PtL_2_@RNPs. Neither cell line exhibited broad absorption due to crystallized PtL_2_, suggesting a lack of decomposition of nanoparticles in the cells [27]. The areas showing the NIR absorption of PtL_2_@RNPs in RGK cells were distributed around the fluorescence of Hoechst 33342, which stained the nuclei (Figure 5d). When RGK1 cells were incubated with PtL_2_@RNPs at 4 °C for 30 min, intracellular NIR absorption of the nanoparticles markedly weakened compared to the incubation at 37 °C for 30 min (Appendix A). Therefore, PtL_2_@RNPs were considered to be internalized by the cells via endocytic pathways. Our group reported that RNPs mostly decompose at pH levels < 6 because of the pH-responsive amino groups in the PMNT chains [29]. Therefore, PtL_2_@RNPs in cells might localize to endosomes with higher pH rather than to lysosomes (pH < 5).

To quantitatively analyze the cellular uptake of PtL_2_@RNPs, we incubated RGM1 and RGK1 cells with PtL_2_@RNPs ([Pt^II^] = 2.0 × 10^−5^ M) for 24 h and measured the Pt concentration of the cell suspensions using ICP-AES (*n* = 3). The intracellular amounts of Pt were (0.32 ± 0.03) × 10^−15^ mol cell^−1^ in RGM1 cells and (0.61 ± 0.04) × 10^−15^ mol cell^−1^ in RGK1 cells. RGK1 cells internalized 1.9 times as much Pt as RGM1 cells (*p* = 0.00082). PtL_2_@RNPs were taken up more efficiently by cancer cells, although they were not modified on the surface with any targeting material specifically recognized by cancer cells.

### 3.5. Cancer Cell-Killing Induced by Photothermal Conversion of PtL_2_@RNPs Under NIR Irradiation

Figure 6 shows the temporal changes in the phase-contrast images of the cells after irradiation by the 730 nm NIR laser (0.15 W) for 10 min. Time-lapse videos of RGM1 and RGK1 cells after treatment are shown in Appendix A. In the RGM1 cells, slight cell shrinkage or rupture was observed in the irradiated areas. In contrast, the RGK1 cells in the laser spot started to shrink significantly after 2 h, and the ambient cells gathered there. This suggests that the photothermal conversion of PtL_2_@RNPs damaged RGK1 cells. When the irradiation power increased (0.28 W), the RGM1 cells shrank. Moreover, in the irradiated area of the RGK1 cells, a blank space was temporarily created by rapid cell shrinkage, which was eventually filled by the surrounding cells Appendix A. Neither cell line was damaged at 0.086 W Appendix A. PtL_2_@RNPs killed cancer cells at a lower irradiation power than normal cells, indicating the potential for cancer-specific PTT. Simultaneously, these imply that the laser irradiation must be controlled within a specific power range for clinical application. One of the reasons for these results could be the difference in the uptake of PtL_2_@RNPs by RGM1 and RGK1 cells. The heat sensitivities of these cell lines may be related to differences in their damage. Our group has reported that RGK36 cells (RGK1-derived sub-clone cell line) produce more mitochondrial ROS through incubation at 42 °C than at 37 °C, whereas ROS production of RGM1 cells is unchanged [40]. It has also been reported that thermal stimulation causes cancer-cell-specific apoptosis-related gene expression in normal and cancer cell lines [41]. Other cellular features such as light sensitivity and repair capability may also be related to the difference in photothermal damage between the two cell lines. Further investigations are required to explain the mechanisms of cell death in PtL_2_@RNPs-based PTT in more detail.

## 4. Conclusions

In this work, aiming to enhance PTT efficacy through ROS scavenging, we developed diradical-platinum(II) complex-loading antioxidant nanoparticles (PtL_2_@RNPs). Herein, we report the physical, spectroscopic, and photothermal conversion properties and the photothermal cell-killing ability of PtL_2_@RNPs. Although the particle sizes of PtL_2_@RNPs were 20–30 nm, their EEs exceeded 90%, and LCs reached up to 12%. Their structures were stable during long-term storage for a month. These results suggest that RNPs are suitable carriers of PtL_2_ to tumor sites based on the EPR effect. PtL_2_@RNPs in PBS exhibited NIR absorption and stable photothermal conversion. The *η* value of PtL_2_@RNPs was comparable to that of PtL_2_, indicating that the photothermal properties of PtL_2_ were retained after encapsulation into RNPs. Moreover, their photothermal conversion was quite efficient compared with that of conventional PTAs. RGM1 (normal) and RGK1 (cancer) cells remained alive after incubation with PtL_2_@RNPs for 24 h. Despite not being modified to recognize cancer cells, PtL_2_@RNPs were internalized into RGK1 cells approximately twice as much as into RGM1 cells. After irradiation with an NIR laser (0.15 W) for 10 min, RGK1 cells containing PtL_2_@RNPs were damaged over time, whereas RGM1 cells remained mostly unchanged. The photothermal conversion of PtL_2_@RNPs specifically kills cancer cells. However, we have not elucidated the mechanisms underlying cancer cell-specific death in our system. With quantitative evaluation of the photothermal cell-killing effect, further investigation is required to determine why PtL_2_@RNPs-based PTT is effective against cancer cells through analyses of the expression of apoptotic genes or heat shock proteins, the involvement of oxidative stress, etc. Eventually, these investigations will provide insights that will contribute to basic studies on cancer cells and the development of effective cancer treatment modalities.

## Figures and Tables

**Figure 1 nanomaterials-15-00796-f001:**
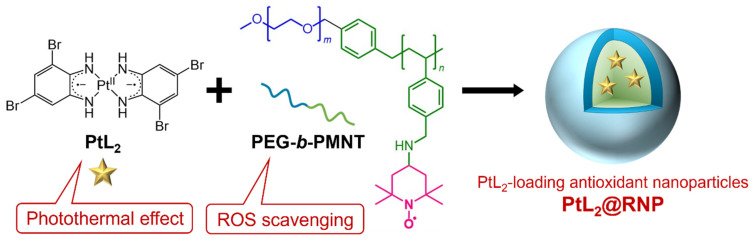
PtL_2_-loading antioxidant nanoparticles PtL_2_@RNP. Blue: polyethylene glycol (PEG) segment, green: hydrophobic PMNT segment. 2,2,6,6-tetramethylpiperidine-N-oxyl (TEMPO) in the PMNT segment is highlighted in magenta.

**Figure 2 nanomaterials-15-00796-f002:**
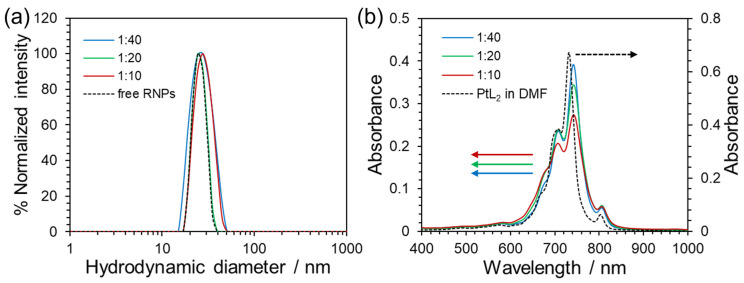
(**a**) Scattering intensity-weighted particle size distributions of the PBS solutions of free RNPs or PtL_2_@RNPs prepared using different PtL_2_/PEG-*b*-PMNT mass ratios. [Pt^II^] = 4 × 10^−5^ M. (**b**) Absorption spectra of the PBS solutions of PtL_2_@RNPs prepared using different PtL_2_/PEG-*b*-PMNT mass ratios (left axis) and PtL_2_ in DMF (right axis). [Pt^II^] = 5.0 × 10^−5^ M (PtL_2_@RNPs in PBS), 6.0 × 10^−5^ M (PtL_2_ in DMF).

**Figure 3 nanomaterials-15-00796-f003:**
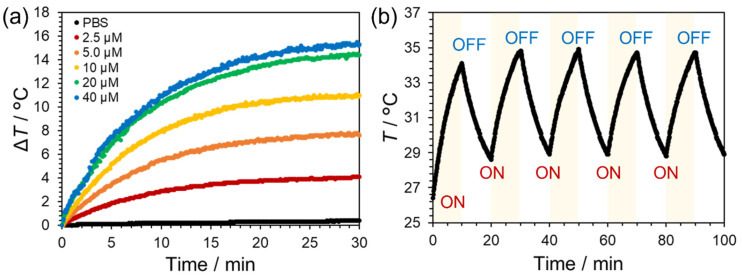
(**a**) Temporal change in solution temperature from the initial value (Δ*T*) of PtL_2_@RNPs in PBS at different Pt^II^ concentrations (0–4.0 × 10^−5^ M) during irradiation by a 730 nm NIR laser (2 W cm^−2^) for 30 min. (**b**) Time course of the PBS solution of PtL_2_@RNPs during five-cycle laser irradiation. [Pt^II^] = 1.0 × 10^−5^ M. In each cycle, the sample was irradiated by a 730 nm NIR laser (2 W cm^−2^) for 10 min and then naturally cooled for 10 min.

**Figure 4 nanomaterials-15-00796-f004:**
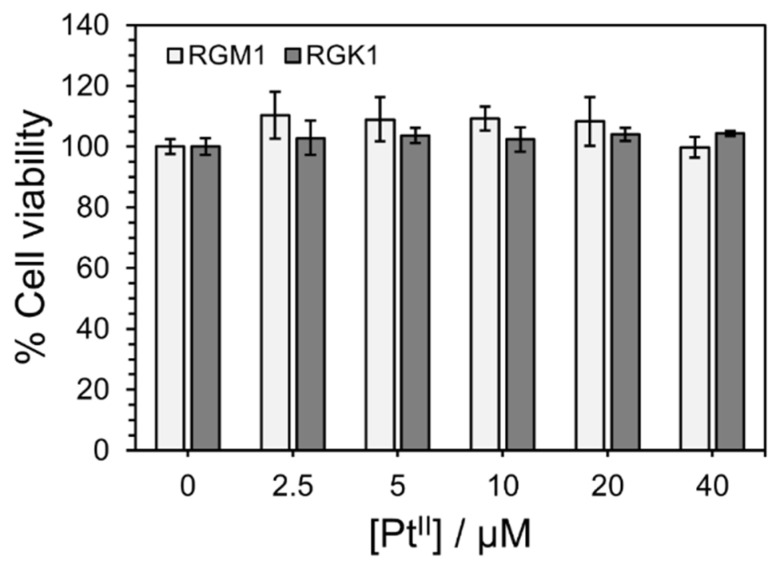
Viabilities of RGM1 and RGK1 cells treated by PtL_2_@RNPs with different Pt^II^ concentrations for 24 h. Error bars represent the standard deviation (*n* = 4).

**Figure 5 nanomaterials-15-00796-f005:**
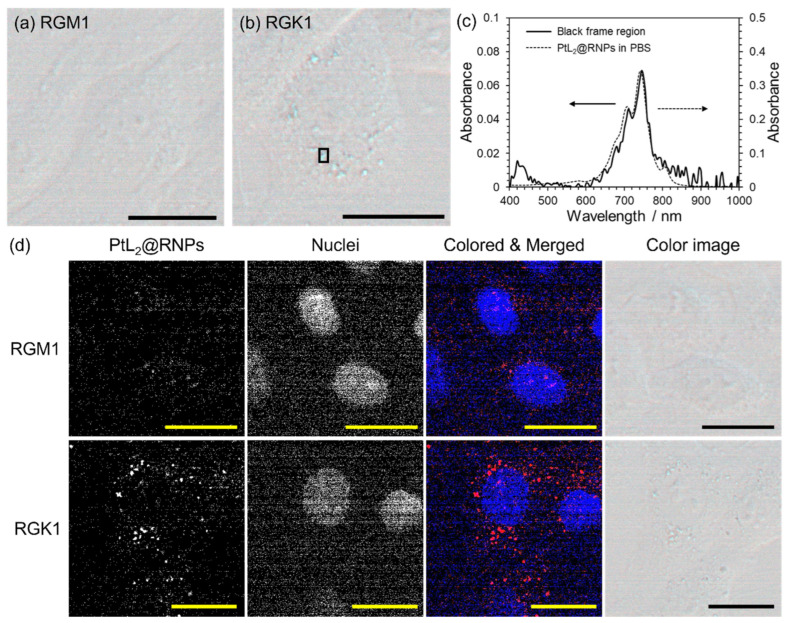
(**a**,**b**) Color images of (**a**) RGM1 and (**b**) RGK1 cells incubated with PtL_2_@RNPs ([Pt^II^] = 4.0 × 10^−5^ M) for 24 h. (**c**) Absorption spectra of black frame region in (**b**) (solid line, left axis) and PtL_2_@RNPs in PBS (dashed line, right axis, [Pt^II^] = 5.0 × 10^−6^ M). (**d**) Subcellular localization analyses of PtL_2_@RNPs in RGM1 and RGK1 cells based on the spectral angle mapper algorithm. From the left, the distribution maps of NIR absorption derived from PtL_2_@RNPs and the fluorescence of Hoechst 33342 (nuclei marker), the merged image of PtL_2_@RNPs (red) and nuclei distribution (blue), and the color image. All scale bars represent 10 µm.

**Figure 6 nanomaterials-15-00796-f006:**
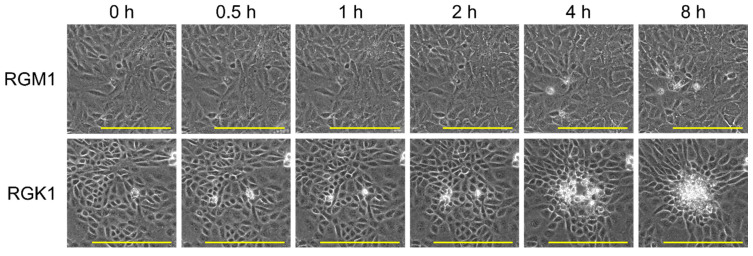
Temporal changes in the phase-contrast images of RGM1 and RGK1 cells containing PtL_2_@RNPs after irradiation by the 730 nm NIR laser (0.15 W, spot size: ~0.3 mm) for 10 min. All scale bars represent 200 µm.

**Table 1 nanomaterials-15-00796-t001:** Physical properties of free RNPs and PtL_2_@RNPs prepared for different PtL_2_/PEG-*b*-PMNT mass ratios. The mean hydrodynamic diameters were calculated from the scattering intensity-weighted size distributions.

Mass Ratio	Mean Hydrodynamic Diameter [nm]	Polydispersity Index	Zeta Potential [mV]	EE [%]	LC [%]
1:40	25.9	0.072	−0.40	99	3.4
1:20	25.8	0.103	0.60	92	5.3
1:10	26.7	0.039	−0.16	91	12
free RNPs	24.9	0.103	−0.94	-	-

## Data Availability

The data presented in this study are included in the article and Appendix A.

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
