# Peer review of "Selective Photothermal Therapy Using Antioxidant Nanoparticles Encapsulating Novel Near-Infrared-Absorbing Platinum(II) Complexes"

_nanomaterials, 2025, doi:10.3390/nano15110796_

Round 1
Reviewer 1 Report
Comments and Suggestions for Authors
This study introduces a novel approach to photothermal therapy (PTT) utilizing PtL2-loaded antioxidant nanoparticles (PtL2@RNPs). The manuscript is well-structured, and the findings indicate promising results in cancer-cell-specific photothermal killing. However, several methodological and interpretive aspects need clarification to enhance the manuscript's impact and validity.
- The reported photothermal conversion efficiency of approximately 103% surpasses the values for gold nanorods (approximately 95%) and other photothermal agents (PTAs). Please clarify the calculation method used (e.g., laser power calibration, heat dissipation corrections) in the main text. Additionally, validate the supplemental data to ensure that there is no overestimation due to experimental artifacts.
- The article indicates that the nanoparticles demonstrate minimal cytotoxicity to cells in the absence of near-infrared (NIR) irradiation. However, it has not been verified whether they induce any cytotoxicity following NIR irradiation.
- The study lacks essential control experiments concerning biological effects, such as a group that is exposed solely to near-infrared (NIR) irradiation for comparison purposes.
- At a power level of 0.28 W, RGM1 cells exhibit damage, which diminishes therapeutic specificity. This raises the importance of discussing the therapeutic window and proposing strategies to optimize laser parameters for clinical application.
Increase the sample size for ICP-AES (from n=3 to n≥5) and cytotoxicity assays to enhance statistical reliability. Additionally, include p-values for all comparative analyses.
Author Response
Thank you very much for taking the time to review this manuscript. Please find the detailed responses below and the corresponding revisions/corrections in track changes in the resubmitted files.
Comments 1: This study introduces a novel approach to photothermal therapy (PTT) utilizing PtL2-loaded antioxidant nanoparticles (PtL2@RNPs). The manuscript is well-structured, and the findings indicate promising results in cancer-cell-specific photothermal killing. However, several methodological and interpretive aspects need clarification to enhance the manuscript's impact and validity.
1. The reported photothermal conversion efficiency of approximately 103% surpasses the values for gold nanorods (approximately 95%) and other photothermal agents (PTAs). Please clarify the calculation method used (e.g., laser power calibration, heat dissipation corrections) in the main text. Additionally, validate the supplemental data to ensure that there is no overestimation due to experimental artifacts.
Response 1: Thank you for pointing this out. We have shown the calculation method of photothermal conversion efficiency (η) in pp. S4–S5 of the Supplementary Information. This method is based on the reported model (Small, 2010, 6, 2272–2280.). The thermal energy balance of the quartz cuvette containing the solution can be described using the heat generation by laser irradiation (Qlaser) and heat dissipation from the system (Qloss). The proportion of heat generation from the solvent and cuvette by laser irradiation is considered as ξ based on measurement using only the solvent. Qloss can be described to be a Taylor series of ΔT containing two unknown coefficients B [W K–1] and C [W K–2]. These parameters are determined by fitting the temperature curve during the cooling period after laser irradiation to the model when Qlaser = 0. Finally, η is calculated from the temperature increase (ΔT) when the temporal change in ΔT becomes 0, and some physical parameters. The laser power was measured by a laser power meter (PM160T-HP, Thorlabs, Inc.).
For the possibility of overestimation, the most conceivable cause is the variation of the output laser power. The used laser had the nominal output stability of <3% per 2 hours. When we calibrated the laser power, we adopted the approximate median considering the variation of the display value of a power meter. The actual output power may have differed from the power when calibrating. We tried to estimate the effect of the error for other parameters (heat capacity, the proportion of heat generation from the solvent and cuvette, and absorbance of the sample) on η, but could not find any parameters that could explain the excess.
Therefore, we have revised the description on the laser power calibration in Section 2.5 (p. 4 in the revised manuscript) as follows: “The laser power (2 W cm–2) was calibrated at an approximate median considering variations in the detected value by the laser power meter (PM160T-HP, Thorlabs, Inc.).”
Moreover, we have added the sentence about the possibility of overestimation to Section 3.2 (p. 8 in the revised manuscript) as follows: “The η value of PtL2@RNPs was calculated as 103% (Figure S5), almost the same as that of PtL2 (Figure S6). A variation in the laser output power might cause a 3% excess of the η value. Gold nanorods used as...”
Comments 2: 2. The article indicates that the nanoparticles demonstrate minimal cytotoxicity to cells in the absence of near-infrared (NIR) irradiation. However, it has not been verified whether they induce any cytotoxicity following NIR irradiation.
Response 2: Thank you for your comment. In a preliminary experiment, we irradiated RGM1 (normal) or RGK1 (cancer) cells, either containing no PtL2@RNPs, with a NIR laser (0.28 W) for 10 min. This irradiation power is the strongest condition adopted in this report. No remarkable changes in these cells were observed. Since this experiment was conducted to set conditions, we did not take time-lapse movies. On the other hand, we had taken phase contrast images of two cell lines immediately and 24 h after laser irradiation (but no image at 24 h for RGM1). These images are shown here. This result at least suggests that NIR laser irradiation alone has no effect on RGK1 cells until 24 h later, and is not acutely toxic to RGM1 cells.

Figure for review only. Phase-contrast images of RGK1 or RGM1 cells immediately and 24 h after laser irradiation (0.28 W, 10 min). There is no image at 24 h for RGM1 cells. Scale bars represent 250 µm.
Comments 3: 3. The study lacks essential control experiments concerning biological effects, such as a group that is exposed solely to near-infrared (NIR) irradiation for comparison purposes.
Response 3: Thank you for pointing this out. As answered in Comments 2, we conducted control experiments to set conditions preliminarily. Although the data is insufficient to compare with PtL2@RNPs-treated conditions, we think that NIR laser irradiation alone does not cause damage to RGM1 and RGK1 cells.
Comments 4: 4. At a power level of 0.28 W, RGM1 cells exhibit damage, which diminishes therapeutic specificity. This raises the importance of discussing the therapeutic window and proposing strategies to optimize laser parameters for clinical application.
Response 4: Thank you for your comment. We also agree with the importance of optimizing laser irradiation parameters for clinical applications.
We have added the sentence about it to Section 3.5 (p. 11 in the revised manuscript) as follows: “PtL2@RNPs killed cancer cells at a lower irradiation power than normal cells, indicating the potential for cancer-specific PTT. Simultaneously, these imply that the laser irradiation must be controlled within a specific power range for clinical application. One of the reasons...”
Comments 5: Increase the sample size for ICP-AES (from n=3 to n≥5) and cytotoxicity assays to enhance statistical reliability. Additionally, include p-values for all comparative analyses.
Response 5: Thank you for your comment. Although we understand that we need to increase the sample size for statistical reliability, time is too short to do additional experiments within the revision deadline. We think the academic significance of this manuscript is the phenomenological reporting of cancer-specific photothermal cell-killing effects based on the time-lapse observation. Designing experimental systems for more detailed quantitative evaluation of the cell-killing effect is currently under consideration.

Reviewer 2 Report
Comments and Suggestions for Authors
This article investigates the synthesis, the material characterization, and the photothermal behavior in PBS and in cells. The authors have used a previously utilized NIR-absorbing hydrophobic diradical-platinum(II) complex (PtL2) to synthesize PtL2-loaded nanoparticles (PtL2@RNPs). PtL2@RNPs present a very high photothermal conversion efficiency. They succeeded in demonstrating a killing effect on cancer cells using their PtL2@RNPs.
The manuscript is well-written and the message is clear. But, I have several minor and major remarks:
- In the abstract the authors have said: “PtL2@RNPs exhibited NIR absorption and effectively killed cancer cells at a low irradiation power (0.15 W).”. The total irradiation power can be very low, but the efficiency of PTT depends more on the power surface density of the laser (usually given in W/cm2) and the shape/volume of the irradiated sample. Therefore, I propose that the authors revise the abstract accordingly.
- A major point concerns the term RNPs. The term PtL2@RNPs is well introduced in the abstract and the introduction, but the reader should deduce that the term RNPs refers to loaded antioxidant nanoparticles. This is not very clear. Furthermore, while it is easy to understand what PtL2 is, there is no clear introduction concerning antioxidant nanoparticles. In the introduction, it is not very clear what the components of these particles are.
- In section “2.4. Preparation of PtL2@RNPs”, if I understood correctly, the nanoparticles are created by mixing PtL2 with the PEG-b-PMNT polymer. For me, the notation “PtL2@RNPs” is misleading. The notation “@” is more commonly used for core@shell objects. In your case, we do not know exactly how the distribution is done, so I believe that PtL2/RNPs is a better notation.
- In my opinion, it is really difficult to discuss nanoparticles without Transmission Electron Microscopy (TEM) images. For me, this is the major revision needed for the article. While dynamic light scattering (DLS) provides information on stability and hydrodynamic size, it does not give access to the actual size and shape of the nanoparticles.
- I really appreciate the effort from the author for the calculation of η. However, the author claims to have a photothermal conversion efficiency of 103%, which is above 100%. The photothermal conversion efficiency coefficient represents the quantity of light energy which is converted into heat. The first law of thermodynamics indicates that you have a conservation of energy, in other words, you cannot create energy. With a coefficient of 103 % the author succeed to create 3% of heat energy. From your extraction of η you should have an error, I hope that your error is higher than 3%…
- Concerning the model in the ESI for extracting the photothermal conversion efficiency, you cannot proceed as you are currently doing. Aλ is the absorbance of the sample at the irradiation wavelength, including the cuvette and solvent or not? In your calculation, it seems that it is the absorbance for only your nanoparticles. Then Qlaser = I (1-10-Aλ)η + I ξ, with (1-10-Aλ) the fraction of light absorbed by the nanoparticles and η the fraction of light energy converted into heat. And, ξ represents the fraction light absorbed and converted into heat by the cuvette and the solvant. And that it is an approximation, because you neglect the diffusion and the inhomogeneity of the temperature of the sample during light irradiation. If Aλ represents the absorbance of the whole system, then Qlaser = I (1-10-Aλ)η. And η will be the value for the whole system. If you think I am wrong, you should make a clearer explanation of your model, and explain ALL your approximations. If not, please do the calculation again.
- I do not think that the way you calculate ξ is correct. To calculate it, you need to perform the photothermal experiment without nanoparticles. Usually ξ tends to 0, it is why we work in the NIR region.
- Concerning the “Cancer cell-killing induced by photothermal conversion” part. The authors claim that some effects are visible on cancer cell for specific low laser power. I agree that 0.15 W can be small, but if you calculate the light power density for a laser beam at a power of 0.15 W and a spot size of 0.3 mm, your power density is around 212 W/cm2… This is 100 times higher compare to your solution temperature study. Is it viable for cancer treatment in vivo?
I believe this article can meets the standards for publication in Nanomaterials after major revision.
Author Response
Thank you very much for taking the time to review this manuscript. Please find the detailed responses below and the corresponding revisions/corrections in track changes in the resubmitted files.
Comments 1: This article investigates the synthesis, the material characterization, and the photothermal behavior in PBS and in cells. The authors have used a previously utilized NIR-absorbing hydrophobic diradical-platinum(II) complex (PtL2) to synthesize PtL2-loaded nanoparticles (PtL2@RNPs). PtL2@RNPs present a very high photothermal conversion efficiency. They succeeded in demonstrating a killing effect on cancer cells using their PtL2@RNPs.
The manuscript is well-written and the message is clear. But, I have several minor and major remarks:
- In the abstract the authors have said: “PtL2@RNPs exhibited NIR absorption and effectively killed cancer cells at a low irradiation power (0.15 W).”. The total irradiation power can be very low, but the efficiency of PTT depends more on the power surface density of the laser (usually given in W/cm2) and the shape/volume of the irradiated sample. Therefore, I propose that the authors revise the abstract accordingly.
Response 1: Thank you for your advice. In the photothermal cell-killing experiment, due to the quite small size of the laser spot, we could not determine the actual diameter of the irradiation area at the cell adhesive surface of a 35-mm film-base dish. Therefore, we avoid describing the irradiation power with the unit of W/cm2.
Comments 2: A major point concerns the term RNPs. The term PtL2@RNPs is well introduced in the abstract and the introduction, but the reader should deduce that the term RNPs refers to loaded antioxidant nanoparticles. This is not very clear. Furthermore, while it is easy to understand what PtL2 is, there is no clear introduction concerning antioxidant nanoparticles. In the introduction, it is not very clear what the components of these particles are.
Response 2: Thank you for pointing this out. RNPs, abbreviation ‘redox nanoparticles’, are nanomicelles composed of the amphiphilic copolymer PEG-b-PMNT. Hydrophobic PMNT segments have 2,2,6,6-tetramethylpiperidine-N-oxyl (TEMPO) moieties, which can scavenge the reactive oxygen species.
Since we did not mentioned the term RNPs, we added its description to Introduction as follows: “The amphiphilic copolymer PEG-b-PMNT consists of a hydrophilic polyethylene glycol (PEG) segment and a hydrophobic PMNT segment, in which 2,2,6,6-tetramethylpiperidine-N-oxyl (TEMPO) moieties are introduced as side chains onto the poly(methylstyrene) backbone via amine linkers [28,29]. This copolymer in an aqueous solution forms nanomicelles, which are named redox nanoparticles (RNPs). When RNPs disintegrate, …”
Comments 3: In section “2.4. Preparation of PtL2@RNPs”, if I understood correctly, the nanoparticles are created by mixing PtL2 with the PEG-b-PMNT polymer. For me, the notation “PtL2@RNPs” is misleading. The notation “@” is more commonly used for core@shell objects. In your case, we do not know exactly how the distribution is done, so I believe that PtL2/RNPs is a better notation.
Response 3: Thank you for your comments. As you mentioned, PtL2@RNPs are prepared by mixing PtL2 and PEG-b-PMNT polymer in DMF, followed by dialysis against water. In the formation of nanomicelles, PtL2 interacts with polymers non-covalently. Specifically, the complex is thought to interact with the aromatic ring in the hydrophobic PMNT segment via π-π stacking. We use the notation “@” to express that PtL2 is encapsulated in RNPs without any covalent bonding. In addition, since we describe the PtL2 and polymer mass ratio as PtL2/PEG-b-PMNT, we use “@” to avoid the reader’s confusion.
Comments 4: In my opinion, it is really difficult to discuss nanoparticles without Transmission Electron Microscopy (TEM) images. For me, this is the major revision needed for the article. While dynamic light scattering (DLS) provides information on stability and hydrodynamic size, it does not give access to the actual size and shape of the nanoparticles.
Response 4: We agree with your comment that the particle size distribution obtained from the DLS measurement does not exhibit the size and shape of the nanoparticles without hydration. On the other hand, we think that the nanoparticles are hydrated in the physiological environment. Since PtL2@RNPs are ‘soft’ nanoparticles because of polymeric micelles, the dried nanoparticles for observation by TEM may shrink or change their shape significantly.
Comments 5: I really appreciate the effort from the author for the calculation of η. However, the author claims to have a photothermal conversion efficiency of 103%, which is above 100%. The photothermal conversion efficiency coefficient represents the quantity of light energy which is converted into heat. The first law of thermodynamics indicates that you have a conservation of energy, in other words, you cannot create energy. With a coefficient of 103 % the author succeed to create 3% of heat energy. From your extraction of η you should have an error, I hope that your error is higher than 3%…
Response 5: Thank you for pointing this out. We think that a 3% excess of η may be caused by the variation of laser output power. We calibrated the output power using a laser power meter (PM160T-HP, Thorlabs, Inc.). The used laser had the nominal output stability of <3% per 2 hours. Therefore, we adopted the approximate median considering the variation of the display value of a power meter. The actual output power during experiments may have differed from the power when calibrating. We tried to estimate the effect of the error for other parameters (heat capacity, the percentage of heat generation from the solvent and cuvette, and absorbance of the sample) on η, but could not find any parameters that could explain the excess.
We have revised the description on the laser power calibration in Section 2.5 (p. 4 in the revised manuscript) as follows: “The laser power (2 W cm–2) was calibrated at an approximate median considering variations in the detected value by the laser power meter (PM160T-HP, Thorlabs, Inc.).”
Moreover, we have added the sentence about the possibility of overestimation to Section 3.2 (p. 8 in the revised manuscript) as follows: “The η value of PtL2@RNPs was calculated as 103% (Figure S5), almost the same as that of PtL2 (Figure S6). A variation in the laser output power might cause a 3% excess of the η value. Gold nanorods used as...”
Comments 6: Concerning the model in the ESI for extracting the photothermal conversion efficiency, you cannot proceed as you are currently doing. Aλ is the absorbance of the sample at the irradiation wavelength, including the cuvette and solvent or not? In your calculation, it seems that it is the absorbance for only your nanoparticles. Then Qlaser = I (1-10-Aλ)η + I ξ, with (1-10-Aλ) the fraction of light absorbed by the nanoparticles and η the fraction of light energy converted into heat. And, ξ represents the fraction light absorbed and converted into heat by the cuvette and the solvant. And that it is an approximation, because you neglect the diffusion and the inhomogeneity of the temperature of the sample during light irradiation. If Aλ represents the absorbance of the whole system, then Qlaser = I (1-10-Aλ)η. And η will be the value for the whole system. If you think I am wrong, you should make a clearer explanation of your model, and explain ALL your approximations. If not, please do the calculation again.
Response 6: Thank you for your question and comments. Before NIR laser irradiation, we measured the absorption spectrum of the sample using a UV-Vis spectrometer. In advance, the baseline was recorded using the sample and reference cuvettes, which contain the solvent only. The absorbances of the solvent and cuvette were canceled. After that, the solvent in the sample cuvette was replaced with PtL2@RNPs in PBS, followed by the absorption spectrum measurement. Therefore, Aλ represents the absorbance of PtL2@RNPs only at the wavelength λ.
The temperature distribution in the sample solution is assumed to be homogeneous by stirring the solution during the laser irradiation. Heat transfer between the solution and the surrounding air is limited to only through the cuvette walls by capping the cuvette entrance with Styrofoam.
Comments 7: I do not think that the way you calculate ξ is correct. To calculate it, you need to perform the photothermal experiment without nanoparticles. Usually ξ tends to 0, it is why we work in the NIR region.
Response 7: In the literature (Small, 2010, 6, 2272–2280.), which refers to the method in our experiments, the ξ value is reported as 0.0411. This resembles our reporting value in the ESI. As we mentioned in the Introduction, water does not show intense NIR absorption. Therefore, the temperature increase without PtL2@RNPs is mainly thought to derive from the radiant heat from the cuvette wall irradiated by a NIR laser.
Comments 8: Concerning the “Cancer cell-killing induced by photothermal conversion” part. The authors claim that some effects are visible on cancer cell for specific low laser power. I agree that 0.15 W can be small, but if you calculate the light power density for a laser beam at a power of 0.15 W and a spot size of 0.3 mm, your power density is around 212 W/cm2… This is 100 times higher compare to your solution temperature study. Is it viable for cancer treatment in vivo?
Response 8: Thank you for your question. In a preliminary experiment, we irradiated RGM1 (normal) or RGK1 (cancer) cells, either containing no PtL2@RNPs, with a NIR laser (0.28 W) for 10 min. This irradiation power is the strongest condition adopted in this report. No remarkable changes in these cells were observed. Since this experiment was conducted to set conditions, we did not take time-lapse movies. On the other hand, we had taken phase contrast images of two cell lines immediately and 24 h after laser irradiation (but no image at 24 h for RGM1). These images are shown here. This result at least suggests that NIR laser irradiation alone has no effect on RGK1 cells until 24 h later, and is not acutely toxic to RGM1 cells.

Figure for review only. Phase-contrast images of RGK1 or RGM1 cells immediately and 24 h after laser irradiation (0.28 W, 10 min). There is no image at 24 h for RGM1 cells. Scale bars represent 250 µm.
On the other hand, as you mentioned, the irradiation power adopted in this study was very high. This may be caused by the lower uptake of nanoparticles by the cells. RGK1 cells contained Pt with the concentration of (0.61 ± 0.04) × 10–15 mol cell–1. However, strategies to reduce the laser power are needed to enhance the cellular uptake of nanoparticles. For example, it will be effective to conjugate the targeting materials to the surface of nanoparticles.
Comments 9: I believe this article can meets the standards for publication in Nanomaterials after major revision.
Response 9: Thank you for your kind comment.

Reviewer 3 Report
Comments and Suggestions for Authors
Based on previous studies, the authors present a new nanoplatform for photothermal cancer therapy. The work is interesting and worth publishing. However, there are a few points that remain unclear.
Figure 2 (b) is not clear. Maybe it would help with colored lines or enlarging a part of the figure in an inset.
In section 3.2. authors comment on the spectral changes of nanomicelles. However, that term is not mentioned anymore along the text. Can nanoparticles considered to be micelles?
Section 3.3. Why are the viabilities of the two cell lines not studied under NIR irradiation?
Author Response
Thank you very much for taking the time to review this manuscript. Please find the detailed responses below and the corresponding revisions/corrections in track changes in the resubmitted files.
Comments 1: Based on previous studies, the authors present a new nanoplatform for photothermal cancer therapy. The work is interesting and worth publishing. However, there are a few points that remain unclear.
Figure 2 (b) is not clear. Maybe it would help with colored lines or enlarging a part of the figure in an inset.
Response 1: Thank you for your advice. We have replaced the graph lines of Figure 2 (p. 7 in the revised manuscript) with the color-differentiated lines as follows:

Figure 2. (a) Scattering intensity-weighted particle size distributions of the PBS solutions of free RNPs or PtL2@RNPs prepared using different PtL2/PEG-b-PMNT mass ratios. [PtII] = 4 × 10–5 M. (b) Absorption spectra of the PBS solutions of PtL2@RNPs prepared using different PtL2/PEG-b-PMNT mass ratios (left axis) and PtL2 in DMF (right axis). [PtII] = 5.0 × 10–5 M (PtL2@RNPs in PBS), 6.0 × 10–5 M (PtL2 in DMF).
Comments 2: In section 3.2. authors comment on the spectral changes of nanomicelles. However, that term is not mentioned anymore along the text. Can nanoparticles considered to be micelles?
Response 2: Yes, RNPs are nanomicelles composed of amphiphilic copolymer PEG-b-PMNT. Prof. Nagasaki’s research group has developed these nanomicelles and named them redox nanoparticles (RNPs).
We have added the description on it to the introduction as follows: “The amphiphilic copolymer PEG-b-PMNT consists of a hydrophilic polyethylene glycol (PEG) segment and a hydrophobic PMNT segment, in which 2,2,6,6-tetramethylpiperidine-N-oxyl (TEMPO) moieties are introduced as side chains onto the poly(methylstyrene) backbone via amine linkers [28,29]. This copolymer in an aqueous solution forms nanomicelles, which are named redox nanoparticles (RNPs). When RNPs disintegrate, …”
Comments 3: Section 3.3. Why are the viabilities of the two cell lines not studied under NIR irradiation?
Response 3: Thank you for your question. We have not yet established experimental systems to evaluate cell viability quantitatively under NIR laser irradiation conditions. Designing the system is under consideration. In this report, we were finally able to observe photothermal damage to the cells adhering to the culture dish by placing an optical fiber in contact with the bottom of the dish. However, the laser spot size was 125 µm in diameter, an optical fiber’s cladding diameter, and too small to calculate the percentage of dead cells in the irradiation area. Therefore, we decided to report the phenomenological results obtained by the time-lapse observation.

Round 2
Reviewer 2 Report
Comments and Suggestions for Authors
I have made several comments on the manuscript, and the authors have taken almost all of them into account. However, in my opinion, there are still two important points that need to be clarified before publication.
Comment 4:
It is possible to study polymeric micelles with a contrast agent in a transmission electron microscope (TEM), and the results are generally reliable. Even in the event of some deformation, this information remains beneficial in terms of gaining insight into the characteristics of NPs. The authors should demonstrate the potential outcomes, which may be positive or negative.
Comment 5:
It is really a problem to write 103% in the article, because this value cannot be reached. What I have asked the author to do is to extract an error from the mathematical model. When the authors fit the different parameters for the model, they should give the error on each parameter from the fit. Then, when they use equation (S5), they can calculate the error on η due to the fit. They have an error on B, C and ξ.
The sentence “A variation in the laser output power might cause a 3% excess of the η value.” is not sufficient.
If the authors want to prove that come from the laser, they have to measure the power of the laser (at the end of the optical path) during the experiment and check the 3 %. But it is strange to have twice the same result in two different experiments if the authors have really an error of 3%.
If there is no clear evidence that this 3% is not coming from the laser or the error in the mathematical model, that means there are too many approximations being made in the model they are using.
Author Response
Thank you very much for taking the time to review this manuscript. Please find the detailed responses below and the corresponding revisions/corrections in track changes in the resubmitted files.
Comments 1: I have made several comments on the manuscript, and the authors have taken almost all of them into account. However, in my opinion, there are still two important points that need to be clarified before publication.
Comment 4:
It is possible to study polymeric micelles with a contrast agent in a transmission electron microscope (TEM), and the results are generally reliable. Even in the event of some deformation, this information remains beneficial in terms of gaining insight into the characteristics of NPs. The authors should demonstrate the potential outcomes, which may be positive or negative.
Response 1: Thank you for your advice. We tried to observe PtL2@RNPs by TEM, but could not obtain adequate contrast. Therefore, we took Z-contrast images of the nanoparticles and added them to the SI as Figure S1. We can see spherical particles with a diameter of ca. 20 nm.

Figure S1. Z-contrast images of PtL2@RNPs. Scale bars represent (a) 300 nm and (b) 200 nm.
We added the information about the electron microscope to section 2.2: “Z-contrast images of PtL2@RNPs were taken by a Hitachi High-Tech HD-2700 scanning transmission electron microscope. The nanoparticles were loaded on a Collodion film-coated TEM grid (Nissin EM Co., Ltd.) and dried overnight before observation. Ultraviolet-visible (UV-Vis) ...”
We added the result to section 3.1: “Z-contrast images of PtL2@RNPs also exhibited spherical particles with a diameter of ca. 20 nm (Figure S1). The zeta potentials ...”
Additionally, we revised the numbering of figures in the SI.
Comments 2:
Comment 5:
It is really a problem to write 103% in the article, because this value cannot be reached. What I have asked the author to do is to extract an error from the mathematical model. When the authors fit the different parameters for the model, they should give the error on each parameter from the fit. Then, when they use equation (S5), they can calculate the error on η due to the fit. They have an error on B, C and ξ.
The sentence “A variation in the laser output power might cause a 3% excess of the η value.” is not sufficient.
If the authors want to prove that come from the laser, they have to measure the power of the laser (at the end of the optical path) during the experiment and check the 3 %. But it is strange to have twice the same result in two different experiments if the authors have really an error of 3%.
If there is no clear evidence that this 3% is not coming from the laser or the error in the mathematical model, that means there are too many approximations being made in the model they are using.
Response 2: Thank you for your comments. We reviewed the calculation model and then found some mistakes. As you pointed out the Qlaser in the first review, we should have considered as Qlaser = I(1–10–Aλ)η + Iξ because the absorbance Aλ is derived from PtL2@RNPs only. Additionally, equation S3, which was solved the differential equation (S2), was wrong. As a result, unknown parameters, B, C, and ξ were recalculated as (2.59 ± 0.02) × 10–2 W K–1, (0.00 ± 3.35) × 10–5 W K–2, and (2.66 ± 0.02) × 10–2, respectively. The average value of C was zero, suggesting that the second-order term in the Taylor series was negligibly small. The η value of PtL2@RNPs in PBS was determined to be 0.999 ± 0.017 (99.9%). A 3% excess of η was a mistake of our calculation, and we consider it was over the potential error coming from the error of parameters. Similarly, the η value of PtL2 in DMSO was recalculated to be 0.954 ± 0.003 (95.4%). We revised Figures S5b and S6b, and the description on the calculation of η in SI.

Figure S5b and S6b. Time course in the cooling period of solution temperature from the initial value (ΔT) of PtL2@RNPs in PBS (left, Figure S5b) and PtL2 in DMSO (right, Figure S6b). The red dashed lines represent the fitting curves.
Last time, we answered that the variation in the laser power might have affected the η value. However, after rethinking, its variation is thought to be negligible because the thermal equilibrium with the ambient air averages it out. Therefore, we deleted the sentence “A variation in the laser output power might cause a 3% excess of the η value.” in Section 3.2 (p.8 in the revised manuscript).

Reviewer 3 Report
Comments and Suggestions for Authors
The authors have responded satisfactorily to the issues raised by the reviewer and the manuscript can be accepted in its present form.
Author Response
Comment: The authors have responded satisfactorily to the issues raised by the reviewer and the manuscript can be accepted in its present form.
Response: We appreciate your review and comment.
Round 3
Reviewer 2 Report
Comments and Suggestions for Authors
The authors have carefully considered all of the comments received. I suggest publishing this article in its current form.